# The transcription factor HLH-26 controls probiotic-mediated protection against intestinal infection through up-regulation of the Wnt/BAR-1 pathway

Yu Sang, Jie Ren, Alejandro Aballay*

Department of Molecular Microbiology and Immunology, Oregon Health & Science University, Portland, Oregon, United States of America

* aballay@ohsu.edu

## Abstract

Probiotics play a critical role in the control of host intestinal microbial balance, protecting the host from gastrointestinal pathogens, modulating the host immune response, and decreasing host susceptibility to infection. To understand the mechanism underlying the protective effect of probiotics against infections through immune regulation, we examined protection against *Salmonella enterica* infection following exposure to nonpathogenic *Enterococcus faecium* in the nematode *Caenorhabditis elegans*. We found that the transcription factor HLH-26, a REF-1 family member of basic helix–loop–helix transcription factors, was required in the intestine for *E. faecium*–mediated protection of *C. elegans* against a lethal *S. enterica* infection. In addition, we uncovered that defense response genes controlled by the canonical Wnt/BAR-1 pathway were activated upon exposure to *E. faecium* in an HLH-26–dependent manner. Our findings highlight a role for REF-1/HLH-26 in the control of the Wnt/BAR-1 pathway in probiotic-mediated protection against gut infection.

## Introduction

The human gut microbiota plays a crucial role in maintaining physiological homeostasis, and increasing evidence indicates that its imbalance has a major impact on health and disease [1]. The impact of the gut microbiota on intestinal homeostasis appears to be so broad that it is believed that the therapeutic and prophylactic effects of some probiotics for a variety of gut-related disorders might be, at least in part, mediated through modification of the microbiota or its function [2]. Probiotic microorganisms exert their effects in a variety of ways, including pathogen defense, host function modulation, and gut barrier integrity, which may ultimately directly or indirectly act against infecting microorganisms [3,4]. Following the administration of probiotics and activation of Toll-like receptors, intestinal epithelial cells or immune cells induce the production of different cytokines and diverse antimicrobial peptides, strengthening the immune system [5]. However, the myriad of mechanisms by which probiotics may modulate immune function to better control microbial infections remain unclear.

**Data Availability Statement:** All data generated or analyzed during this study are included in the manuscript and supporting files. The accession

number for the RNA seq data reported in this paper is Gene Expression Omnibus (GEO: GSE178099).

**Funding:** This work was supported by National Institutes of Health grants GM070977 (AA) and AI156900 (AA). The funders had no role in study design, data collection and analysis, decision to publish, or preparation of the manuscript.

**Competing interests:** The authors have declared that no competing interests exist.

**Abbreviations:** BHI, brain–heart infusion; CFU, colony-forming unit; DAVID, database for annotation, visualization, and integrated discovery; dsRNA, double-stranded RNA; GO, gene ontology; IPTG, isopropyl-β-D-thiogalactoside; LB, Luria–Bertani; NGM, nematode growth medium; qRT-PCR, quantitative reverse transcription PCR; RNAi, RNAi interference; RNAseq, RNA sequencing; SK, slow-killing; TF, transcription factor.

Canonical Wnt signaling, also known as β-catenin–dependent signaling, is highly conserved across species where it contributes to cell cycle control, cytoskeleton reorganization during phagocytosis and cell migration, autophagy, apoptosis, and a variety of inflammation-related events [6–8]. In the off state of the pathway, the β-catenin destruction complex facilitates the phosphorylation of β-catenin by GSK3β, which induces ubiquitination of β-catenin and subsequent proteasomal degradation. When Wnt ligand binds to a Frizzled family receptor, the destruction complex is inhibited, freeing β-catenin from degradation. Accumulation of the cytoplasmic pool of β-catenin induces its translocation into the nucleus where it binds with T-cell factor transcription factor at the Wnt response element DNA sequence and activates the transcription of target genes [7].

To provide insights into new mechanisms underlying responses to probiotics that modulate immune function and regulate pathogen defense, we used the nematode *Caenorhabditis elegans*, which possesses conserved signaling pathways that are involved in the regulation of host–microbe interactions [9–12].

We found that HLH-26, a REF-1 family member of basic helix–loop–helix transcription factors, is required for the protection that a short prophylactic exposure to a nonpathogenic strain of *Enterococcus faecium* confers against a lethal *Salmonella enterica* infection in *C. elegans*. Furthermore, we demonstrated that the Wnt/BAR-1 pathway acts downstream of HLH-26 following exposure to *E. faecium*. The mRNA levels of *cwn-2* (encoding the Wnt legend CWN-2), *bar-1* (encoding the β-catenin homolog BAR-1), *mig-1* (encoding the Frizzled family receptor MIG-1), and downstream effectors as well as the protein level of BAR-1 increased after *E. faecium* treatment in an *hlh-26*–dependent manner. Our findings characterize the REF-1–like transcription factor HLH-26 as a key regulator of the Wnt pathway in probiotic-mediated protection against a lethal gut infection.

## Results

### *hlh-26* is required for probiotic protection against *S. enterica* infection

To explore host mechanisms activated by probiotic exposure that may confer protection against bacterial infections, we first studied 399 genes that are up-regulated and down-regulated in *C. elegans* exposed to *E. faecium* compared with heat-killed *Escherichia coli* [13]. Using WormNet (http://www.functionalnet.org/wormnet/) [14], we found that out of the 399 genes that are differentially expressed in *C. elegans* exposed to *E. faecium* compared with heat-killed *E. coli*, 266 are coexpressed. This set of 266 coexpressed genes corresponds to 125 coexpressed gene groups, which share the same transcription factor binding site and similar expression patterns (S1 Table). To identify potential transcription factors involved in the control of the 125 coexpressed gene groups, we used Regulatory Sequence Analysis Tools (http://rsat.sb-roscoff.fr) [15] to predict common binding motifs for the top 30 coexpressed groups that together represent 79% of all coexpressed genes. This analysis identified 158 common motifs (S2 Table), and the subsequent alignment of those motifs with the reported motifs of 292 *C. elegans* transcription factors (TFs) [16] revealed 28 candidate TFs (S3 Table). Overall, 82 out of the 399 differentially expressed genes are predicted to be regulated by those 28 TFs, and 63 genes are up-regulated while 19 genes are down-regulated (S3 Table). The 5 top TFs are predicted to control 43 genes, 31 up-regulated and 12 down-regulated, of the original 399 differentially expressed genes (S3 Table).

To address whether the TFs potentially activated by *E. faecium* exposure played a role in protecting *C. elegans* from a lethal gut infection, we used a short prophylactic exposure to live *E. faecium*, which is sufficient to protect *C. elegans* against *S. enterica* infection [17]. We also tested whether heat-killed *E. faecium* may also confer protection against *S. enterica* infection

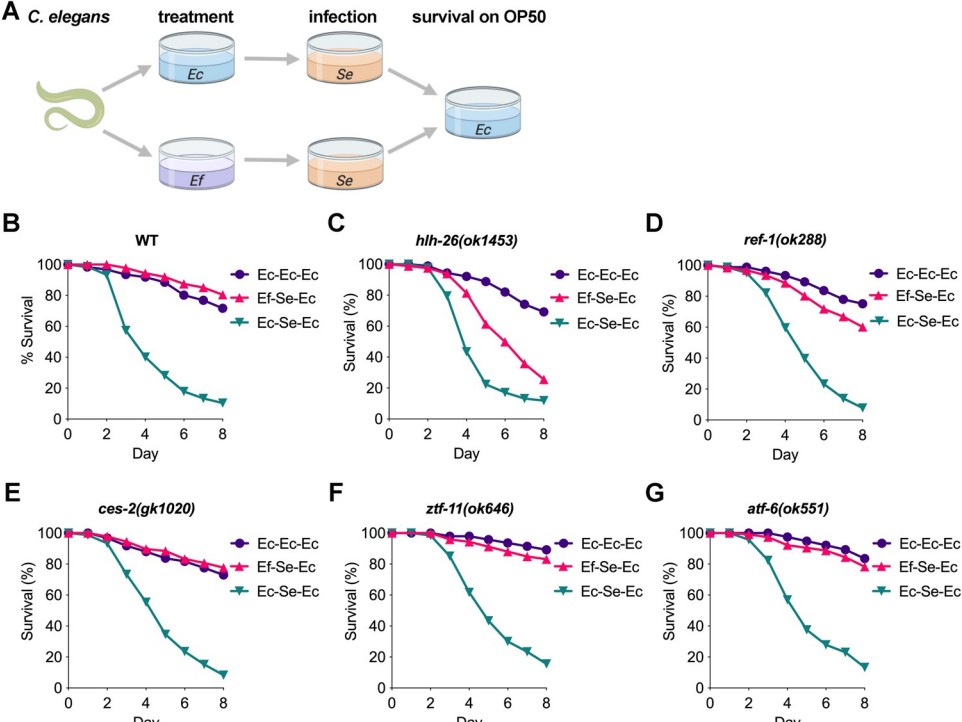

**Fig 1. *E. faecium* protection in wild-type and mutant *C. elegans*.** (A) Schematic of the treatment–1-day infection assay. Worms were treated on *E. faecium* lawns for 1 day before infection with *S. enterica* for 1 day. Worms were then transferred onto *E. coli* OP50 plates for the remainder of the assay, and survival was scored. (B) Survival curve from a treatment–1-day infection assay showing that *E. faecium* inhibits *S. enterica* pathogenesis. Survival curve showing *E. faecium*–mediated protection in (C) *hlh-26(ok1453)*, (D) *ref-1(ok288)*, (E) *ces-2(gk1020)*, (F) *ztf-11(ok646)*, and (G) *atf-6(ok551)*. Survival curves are representative assays of 3 independent experiments. *n* = 60 to 90. The data underlying all the graphs shown in the figure can be found in S1 Data. Ec, *E. coli* OP50; Ef, *E. faecium*; Se, *S. enterica*; WT, wild-type.

and found that the protection was not as strong as that of live *E. faecium* (S1 Fig), indicating that live probiotic is required for full protection. Thus, using live probiotic treatment followed by a 1-day pathogen exposure assay (Fig 1A and 1B), we studied the 5 top-ranked matching TFs (Table 1) and found that the survival of *hlh-26*(*ok1453*) and *ref-1*(*ok288*) animals fed *E. faecium* before infection decreased (Fig 1C and 1D). In contrast, no changes were observed in

**Table 1. Top-ranked candidate TFs.**

| Gene Name | Sequence | Expression location | Binding motifs |
|---|---|---|---|
| *hlh-26* | C17C3.8 | unknown | RACACGTGY RCACGTGTY |
| *ref-1* | T01E8.2 | intestine | SCACGTGK MCACGTGS |
| *ces-2* | ZK909.4 | excretory duct cell | RTTACGYAAY RTTRCGTAAY |
| *ztf-11* | F52F12.6 | g1 and head neurons | VAAGTTTN NAAACTTB |
| *atf-6* | F45E6.2 | intestine; muscle cell; pharyngeal gland cell; rectum; and vulva | NYKACGTDD HHACGTMRN |

TF, transcription factor.

the survival of *ces-2(gk1020)*, *ztf-11(ok646)*, or *atf-6(ok551)* animals fed *E. faecium* before infection (Fig 1E–1G). We found that the survival of *hlh-26(ok1453);ref-1(ok288)* animals was not different than that of *hlh-26(ok1453)* animals fed *E. faecium* before infection, indicating that *hlh-26* and *ref-1* are part of the same pathway (S2 Fig). Because of the specificity of the binding motif (S3 Table) and strong protective phenotype, we decided to focus on HLH-26.

Because the genes we used to predict potential transcription factors required for protection against infection were differentially expressed under *E. faecium* compared with heat-killed *E. coli* exposure, we tested whether HLH-26 was also required for protection against *S. enterica* infection when animals were preexposed to *E. faecium* or heat-killed *E. coli* (S3 Fig). We found that unlike exposure to *E. faecium*, exposure to heat-killed *E. coli* did not confer protection against *S. enterica* infection (S3 Fig). As expected, the survival of *hlh-26(ok1453)* animals did not improve as much as that of wild-type animals pretreated with *E. faecium* (S3 Fig).

To confirm the protective role of HLH-26, we also followed the probiotic pretreatment with a continuous infection assay (Fig 2A). *C. elegans* survival during continuous *S. enterica*

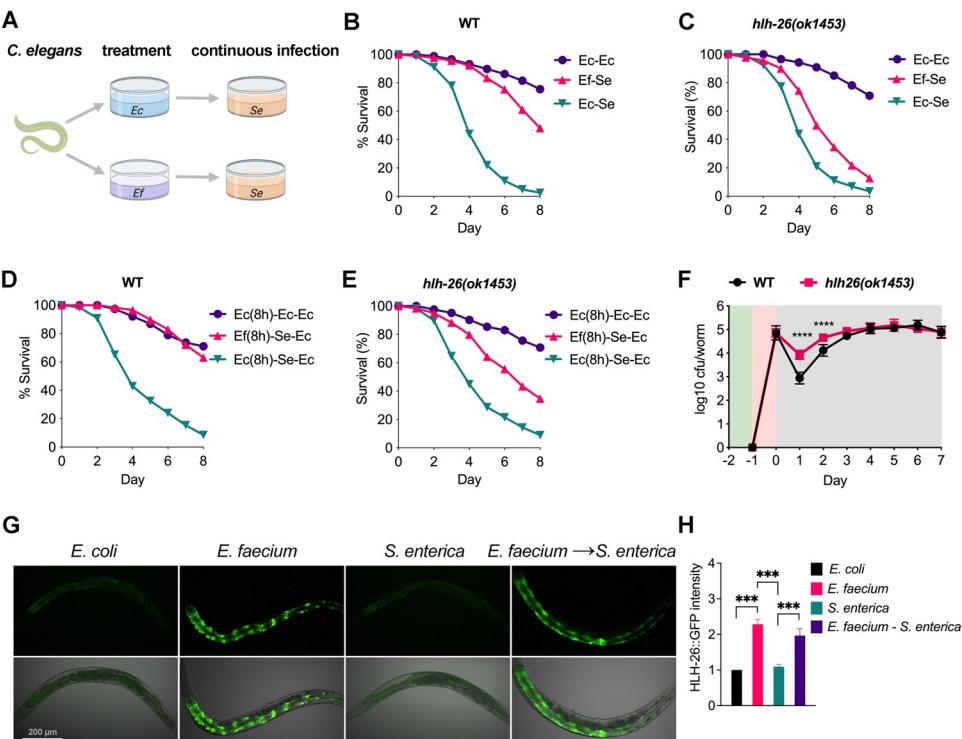

**Fig 2. *hlh-26* is required for probiotic *E. faecium*–mediated protection against *S. enterica* infection.** (A) Schematic of a treatment-continuous *S. enterica* infection assay. Animals were treated for 1 day on BHI agar with *E. faecium* and then transferred to modified NGM agar (0.35% peptone instead of 0.25% peptone) for infection with *S. enterica* for the remainder of the assay, after which survival was scored. Survival curves assaying *E. faecium*–mediated protection with continuous *S. enterica* infection in (B) wild-type and (C) *hlh-26(ok1453)* animals. Survival curves assaying 8 h of *E. faecium* protection in (D) wild-type and (E) *hlh-26(ok1453)* animals. Survival curves are representative assays of 3 independent experiments. *n* = 60 to 90. (F) *S. enterica* CFU measured after infection in wild-type and *hlh-26(ok1453)* animals. Data points represent average CFU ± SD. *n* = 6 animals; *N* = 3 biological replicates. ****$P < 0.0001$. The background shading represents the stage of the treatment–infection assay. Green indicates treatment, red indicates infection, and gray indicates *E. coli* OP50 feeding. (G) Fluorescence micrographs of animals expressing GFP fused to HLH-26 (HLH-26::GFP) after an 8-h exposure to *E. coli*, *E. faecium*, *S. enterica*, or 24-h exposure to *E. faecium* followed by 8-h exposure to *S. enterica*. (H) Quantification of the animals shown in (G), *N* = 3 biological replicates. ***$P < 0.001$. The data underlying all the graphs shown in the figure can be found in S1 Data. CFU, colony-forming unit; Ec, *E. coli* OP50; Ef, *E. faecium*; kEc, heat-killed *E. coli* OP50; NGM, nematode growth medium; Se, *S. enterica*; WT, wild-type.

infection was improved in animals pretreated with *E. faecium* as compared with animals fed *E. coli* (Fig 2B). Survival during *S. enterica* infection of *hlh-26*(*ok1453*) animals was also shorter than that of wild-type animals pretreated with *E. faecium* (Fig 2C). We further found that exposure to *E. faecium* for only 8 h was sufficient for protection against *S. enterica* infection and that *hlh-26* was required for protection under these conditions (Fig 2D and 2E). Viable *S. enterica* colony-forming units (CFUs) recovered from lysed animals revealed an approximately 10-fold increase of *S. enterica* colonization in *hlh-26*(*ok1453)* animals compared to wild-type animals after 1 day of infection (Fig 2F). Moreover, HLH-26 protein level increased and HLH-26 accumulated in nuclei of animals treated with *E. faecium* (Fig 2G and 2H).

Because SagA is an *E. faecium*–secreted peptidoglycan hydrolase previously shown to control the protection of *C. elegans* against *S. enterica* [17], we tested whether HLH-26 was also required for protection against *S. enterica* infection triggered by SagA. We found that the survival after *S. enterica* infection of *hlh-26*(*ok1453*) animals was also shorter than that of wild-type animals pretreated with *E. coli* BL21 expressing SagA (S4 Fig). Just like exposure to *E. faecium* (Fig 2G and 2H), exposure to *E. coli* BL21 expressing SagA increased HLH-26 expression and nuclear accumulation (S4D and S4E Fig). Taken together, these results indicate that HLH-26 is required for *E. faecium*–mediated protection against *S. enterica* infection.

HLH-26 may specifically function in protection against *S. enterica* infection elicited by probiotic exposure, or it may be required for survival against the pathogen. To distinguish between these possibilities, we compared the survival of *hlh-26*(*ok1453*) animals and wild-type animals exposed to *E. faecium* or *S. enterica*. HLH-26 did not seem to be required for *C. elegans* survival in the presence of either *E. faecium* or *S. enterica* (S5 Fig).

We used quantitative reverse transcription PCR (qRT-PCR) to validate the coexpressed groups that were predicted to be regulated by HLH-26. We focused on 3 groups sharing conserved DNA binding motifs for HLH-26 (S3 Table) that together represented a total of 70 genes (S1 Table, groups 2, 3, and 4). Ten genes randomly selected using 4 conditions were verified: wild-type animals exposed to heat-killed *E. coli*, wild-type animals exposed to *E. faecium*, *hlh-26*(*ok1453*) animals exposed to heat-killed *E. coli*, and *hlh-26*(*ok1453*) animals exposed to *E. faecium*. The expression of 7 of 10 genes activated by *E. faecium* was found to be *hlh-26* dependent (S6 Fig). Because *hlh-26* mutation seemed to prevent *E. faecium* activation but not the basal expression of those genes, other transcription factors might be required for their expression. These results indicated that part of *E. faecium*–induced genes were transcriptionally regulated by HLH-26.

## HLH-26 functions in the intestine to protect against *S. enterica* infection upon probiotic exposure

To identify the specific tissues in which *hlh-26* is required for protection against *S. enterica* infection, we generated animals expressing GFP under the control of the *hlh-26* promoter. Fluorescence imaging of *hlh-26p*::*gfp* animals revealed high expression of GFP in the intestinal cells of the animals (Fig 3A), indicating that HLH-26 might function in the intestine. The intestinal function of HLH-26 in protection against *S. enterica* infection was also suggested using strain MGH171, which allows intestine-specific RNAi interference (RNAi). We found that *hlh-26* RNAi in the intestine significantly decreased the *E. faecium*–mediated protection against *S. enterica* infection (Fig 3B and 3C). Consistent with the intestinal expression of HLH-26, we did not observe changes in *E. faecium*–mediated protection between control animals and animals in which *hlh-26* was inhibited by RNAi in either the germline or the nervous system (S7 Fig). Further substantiating the notion that HLH-26 functions in the intestine, HLH-26 expression under the regulation of the intestine-specific promoter *Pvha-6* also fully rescued the mutant phenotype of *hlh-26*(*ok1453*) animals (Fig 3D–3G).

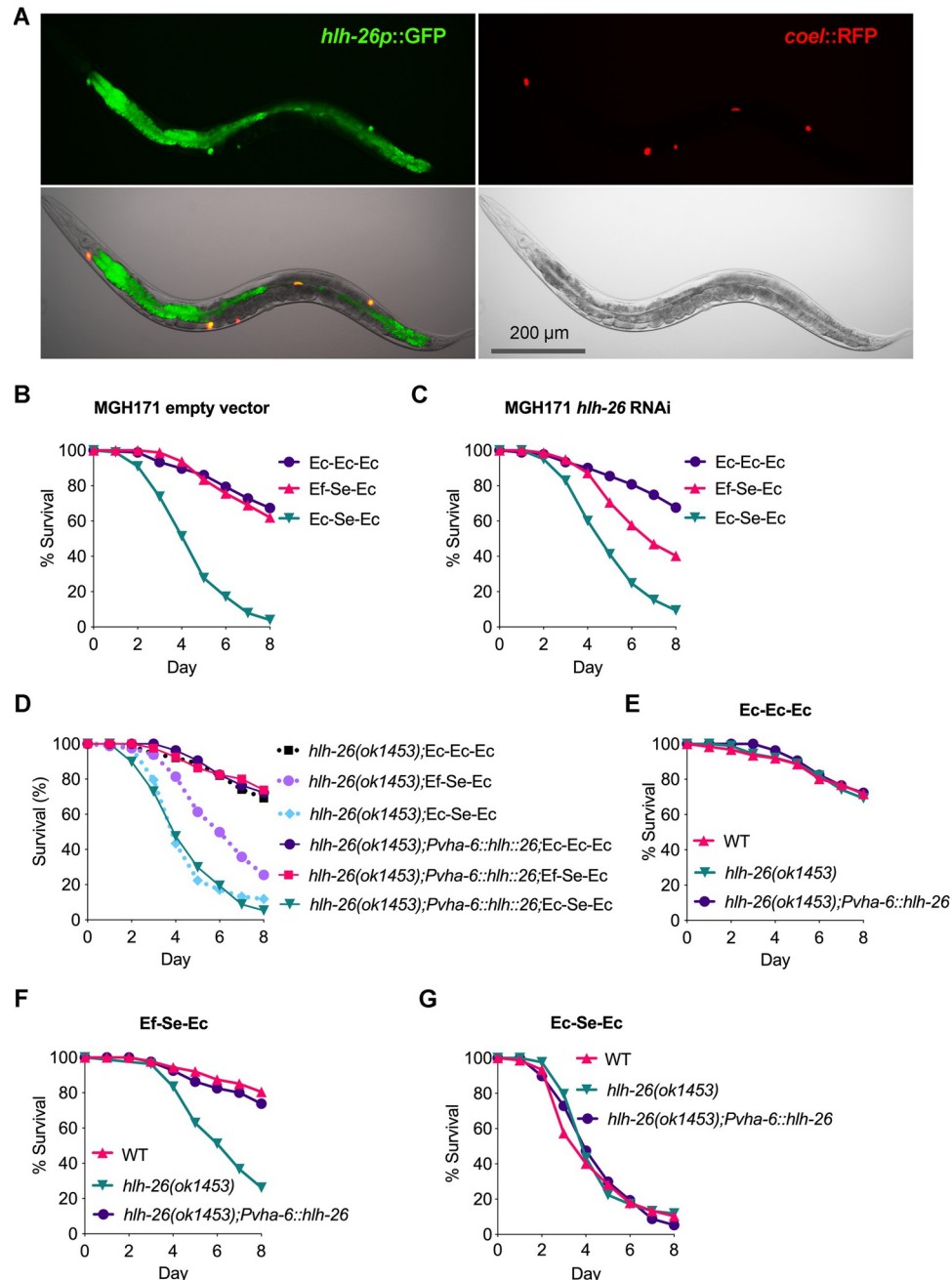

**Fig 3. HLH-26 is expressed in the intestine.** (A) *hlh-26p*::*gfp* expression in young adult transgenic worms assessed by fluorescence microscopy (25×). *coel*::*rfp* was used as the coinjection marker. Survival curves assaying *E. faecium*–mediated protection in the intestine-specific RNAi strain MGH171 feeding on HT115 expressing (B) empty vector or (C) *hlh-26* RNAi animals. (D) Comparison of *E. faecium*–mediated protection between *hlh-26(ok1453)* animals and *hlh-26(ok1453); Pvha-6::hlh-26* rescue animals. *hlh-26(ok1453)*; Ec-Ec-Ec versus *hlh-26(ok1453); Pvha-6::hlh-26*;Ec-Ec-Ec, P = NS; *hlh-26(ok1453)*; Ef-Se-Ec versus *hlh-26(ok1453);Pvha-6::hlh-26*; Ef-Se-Ec, P < 0.0001; *hlh-26(ok1453)*; Ec-Se-Ec versus *hlh-26(ok1453);Pvha-6::hlh-26*;Ec-Se-Ec, P = NS. (E) WT, *hlh-26(ok1453)*, and *hlh-26(ok1453); Pvha-6::hlh-26* animals were grown with *E. coli-E. coli-E. coli* treatment and scored for survival. WT animals versus *hlh-26(ok1453)*, P = NS; *hlh-26(ok1453); Pvha-6::hlh-26*, P = NS. (F) WT, *hlh-26(ok1453)*, and *hlh-26(ok1453); Pvha-6::hlh-26* animals were grown with *E. faecium-S. enterica-E. coli* treatment and scored for survival. WT animals versus *hlh-26(ok1453)*, P = NS; *hlh-26(ok1453); Pvha-6::hlh-26*, P < 0.0001. *hlh-26(ok1453)* animals versus *hlh-26(ok1453); Pvha-6::hlh26*, P < 0.0001. (G) WT, *hlh-26(ok1453)*, and *hlh-26(ok1453); Pvha-6::hlh-26* animals were grown with *E. coli-S. enterica-E. coli* treatment and scored for survival. WT animals versus *hlh-26(ok1453)*, P = NS; *hlh-26(ok1453); Pvha-6::hlh-26*, P = NS. The data underlying all the graphs shown in the figure can be found in S1 Data. NS, nonsignificant; RNAi, RNAi interference; WT, wild-type.

## HLH-26 is required for induction of immune genes in response to *E. faecium* exposure

To understand the mechanism by which HLH-26 regulated protection against *S. enterica* infection after exposure to *E. faecium*, we used RNA sequencing (RNAseq) to focus on transcriptional changes induced by *E. faecium* in *hlh-26(ok1453)* compared with wild-type animals. Of the 3,261 differentially regulated genes, 2,089 genes were up-regulated while 1,172 genes were down-regulated (S4 Table). Of the 3,261 differentially regulated genes, 543 share HLH-26 binding motifs (S4 Table). Further analysis of those 543 genes indicates that 183 are down-regulated in *hlh-26(ok1453)* animals, while 360 are up-regulated in *hlh-26(ok1453)* animals exposed to *E. faecium* (S4 Table), suggesting that HLH-26 may act as both repressor and activator of gene expression. To identify related gene groups directly or indirectly transcriptionally controlled by HLH-26 following *E. faecium* exposure, we performed an unbiased gene enrichment analysis using the database for annotation, visualization, and integrated discovery (DAVID; http://david.abcc.ncifcrf.gov/) [18]. The 10 gene ontology (GO) clusters with the highest DAVID enrichment score are shown in Fig 4A and S5 Table. For the up-regulated gene set, the highest scoring ontology clusters were metabolic process and protein phosphorylation. Other up-regulated clusters included genes associated with transport, ion transport, proteolysis, and signal transduction, among others. For the subset of down-regulated genes that responded to *E. faecium* exposure without *hlh-26*, transcription regulation cluster was the most highly enriched, followed by immune response cluster. Other down-regulated clusters included genes associated with oxidation–reduction process, chemical stimulus sensing, and nucleosome, among others. As expected, a similar enrichment was also observed using a WormBase enrichment analysis tool (https://wormbase.org/tools/enrichment/tea/tea.cgi) [19,20] that is specific for *C. elegans* gene data analyses (S8 Fig). Metabolic process and protein modification were the 2 most enriched up-regulated clusters. Transcription and immune genes were also highly enriched among the 1,172 HLH-26 down-regulated genes (S8 Fig). Of the 31 down-regulated immune genes, only 4 (*ilys-2*, *spp-2*, *cnc-2*, and *clec-165*) are predicted to be direct targets of HLH-26 (S6 Table), suggesting that the control of immune genes by HLH-26 is primarily indirect.

Because individual samples were used for RNAseq, which can provide false candidate genes, we verified by qRT-PCR the *hlh-26*-dependent regulation of 10 genes involved in immune response (S6 Table) under 4 conditions: wild-type animals exposed to *E. coli*, wild-type animals exposed to *E. faecium*, *hlh-26(ok1453)* animals exposed to *E. coli*, and *hlh-26 (ok1453)* animals exposed to *E. faecium*. Ten tested genes were up-regulated upon exposure to *E. faecium* in an *hlh-26*–dependent manner (Fig 4B). Because HLH-26 also affected the basal expression of *C04G6.5* and *cnc-11* (Fig 4B), it is unclear to what extent the reduced expression of these 2 genes in *hlh-26(ok1453)* animals exposed to *E. faecium* is due to the role of HLH-26 on their expression under control conditions or in the presence of *E. faecium*. The reporter strain CB6710 [21], which expresses GFP under the *ilys-3* promoter, was also induced upon *E. faecium* exposure, and the induction was blocked by *hlh-26* RNAi (Fig 4C and 4D). In general, there was a positive correlation between RNAseq and qRT-PCR expression values, and immune genes were activated following exposure to *E. faecium* in an *hlh-26*–dependent manner.

## The canonical Wnt/BAR-1 pathway is required for *E. faecium*–activated protection

To identify potential immune pathways that might play a role in *E. faecium*–mediated protection against *S. enterica* infection, we performed a gene enrichment analysis using the *hlh-26*–

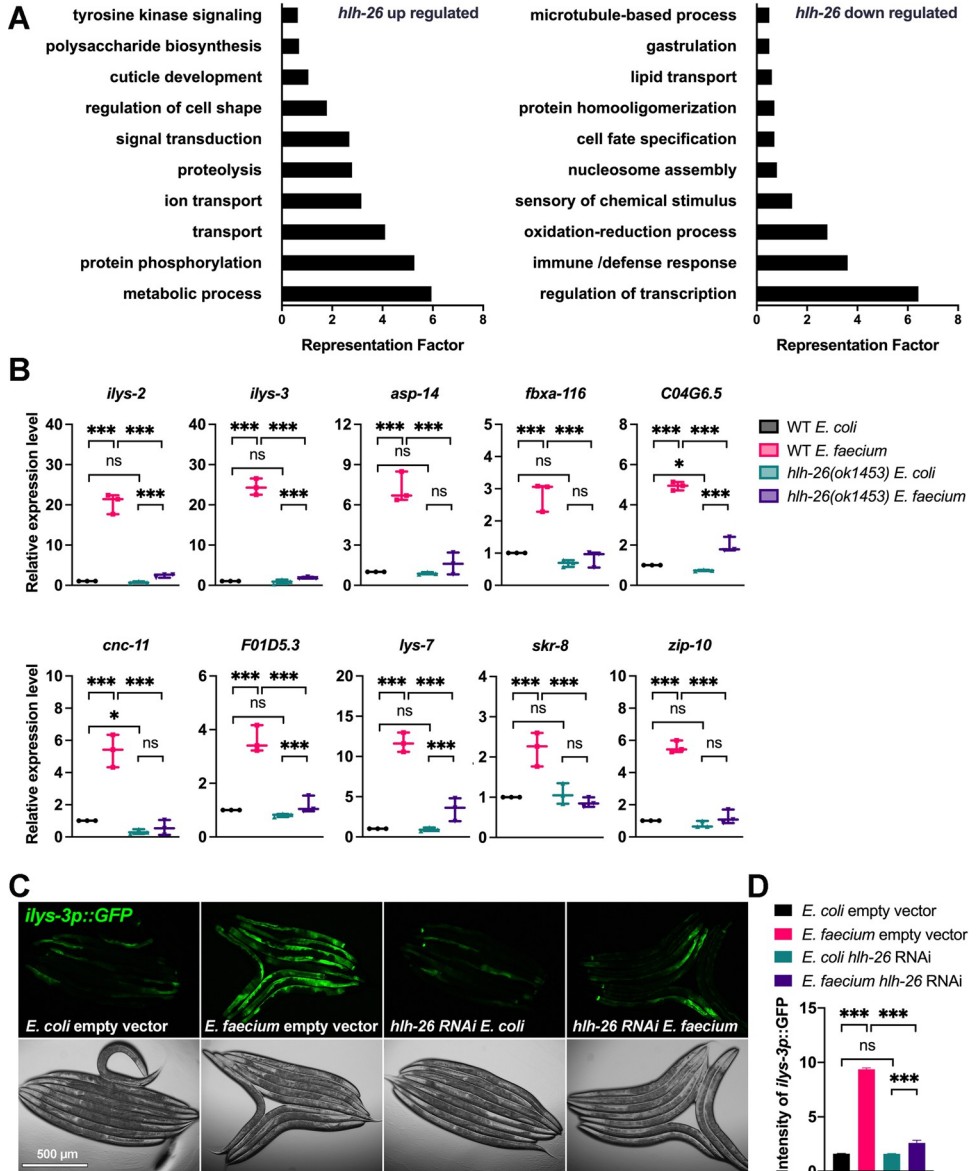

**Fig 4. HLH-26 regulates an innate immune transcriptional program.** (A) GO analysis of up-regulated and down-regulated genes in *hlh-26(ok1453)* animals grown on *E. faecium*. (B) qRT-PCR validation of selected innate immune genes that were down-regulated in *hlh-26(ok1453)* mutants (*N* = 3 biological replicates). Values are expressed as the fold difference compared with wild-type worms fed on *E. coli* ± SD by one-way ANOVA with Tukey's multiple comparisons test, $^*P < 0.05$, $^{**}P < 0.01$, $^{***}P < 0.001$. (C) Fluorescence micrographs of control or *hlh-26* RNAi animals expressing GFP under the *ilys-3* promoter (*ilys-3p::gfp*) after an 8-h exposure to *E. faecium* or *E. coli*. (D) Quantification of the animals shown in (C), *n* = 8. The "n" represents the number of animals in each experiment. *N* = 3 biological replicates. The data underlying all the graphs shown in the figure can be found in S1 Data. GO, gene ontology; ns, nonsignificant; qRT-PCR, quantitative reverse transcription PCR; RNAi, RNAi interference; WT, wild-type.

regulated genes with changes in expression upon exposure to *E. faecium*. Using WormExp (https://wormexp.zoologie.uni-kiel.de/wormexp/), which is a web-based application for gene set enrichment analysis specific for *C. elegans*, we found that several *hlh-26*–regulated genes were also controlled by several other pathways (S7 Table). We then studied *E. faecium*–mediated protection against *S. enterica* infection of *C. elegans* mutants by examining genes

corresponding to the 5 top-ranked pathways as well as 2 immune-related pathways. We found no major roles for the *alg-3/alg-4*, *rsd-2*, *lin-28*, WAGO 22G-RNA, *hlh-30*, or *dbl-1* pathways (S9 Fig). However, *bar-1* was required for *E. faecium*–mediated protection against *S. enterica* infection (Fig 5A), suggesting that *E. faecium* might enhance defense against bacterial infections through activation of canonical Wnt/BAR-1 signaling.

Because *bar-1*(*ga80*) animals were susceptible to bacterial pathogens and even to live *E. coli* (Fig 5A and 5B), we reasoned that animals with a compromised healthspan or susceptible to bacterial infections in general might also be incapable of mounting a protective response against *S. enterica* by pretreatment with *E. faecium*. However, we found that while *fshr-1(ok778)* animals, which are deficient in an intestinal G protein–coupled receptor required for innate immunity, were susceptible to *S. enterica* infection, they were not deficient in *E. faecium*–mediated protection against *S. enterica* infection (Fig 5C and 5D, S10 Fig). These results suggested that BAR-1 was required in *E. faecium*–mediated protection and that the Wnt/BAR-1 pathway might play a critical role in this process. Because BAR-1 showed the highest expression in the posterior intestine, we tested if the intestinal specific RNAi against *bar-1* would reduce the survival of *E. faecium*–treated animals. We found that *bar-1* RNAi in the intestine significantly decreased the *E. faecium*–mediated protection against *S. enterica* infection (Fig 5E and 5F).

To further confirm that Wnt/BAR-1 signaling played an important role during probiotic protection against *S. enterica* infection, we studied *cwn-2*, which encodes a Wnt ligand. Survival during *S. enterica* infection of *cwn-2*(*ok895*) animals pretreated with *E. faecium* was lower than that of wild-type animals (Fig 5G and 5H), while CWN-2 did not seem to be required for *C. elegans* survival in the presence of *S. enterica* (S11 Fig), indicating that the Wnt/BAR-1 pathway was required for *E. faecium* protection against *S. enterica*. We found that *bar-1*, *cwn-2*, and *mig-1*, which encodes the Wnt and receptor Frizzled, were activated by *E. faecium* exposure in an HLH-26–dependent manner (Fig 6A). HLH-26 is also required for basal expression of *cwn-2* (Fig 6A). The expression level of *pop-1*, which encodes a Wnt effector that together with BAR-1 regulates gene expression, was not affected by *E. faecium* exposure or the absence of *hlh-26* (Fig 6A). However, we cannot rule out an effect on the posttranslational activation of POP-1. We further studied Wnt/BAR-1 downstream target genes [22–24] that overlapped with genes that were down-regulated in *hlh-26* animals exposed to *E. faecium* (S8 Table). As shown in Fig 6B, *bli-1*, *his-11*, *oac-30*, *R02E4.1*, *oac-54*, and *fbxa-116* in Fig 4B, which are Wnt/BAR-1 downstream targets [22,24], were activated upon exposure to *E. faecium* in an *hlh-26*–dependent manner.

The Wnt pathway controls the expression of specific target genes through the effector protein β-catenin BAR-1, which is stabilized by the binding of Wnt to the receptor Frizzled [25,26]. To address whether *E. faecium* exposure results in higher levels of BAR-1 in an HLH-26–dependent manner, we detected BAR-1 protein levels using a strain expressing BAR-1::GFP [27]. We found that the BAR-1::GFP signal increased with *E. faecium* exposure and that RNAi inhibition of *hlh-26* resulted in a significant reduction of fluorescence (Fig 6C and 6D). Large particle flow cytometry confirmed the increase in the BAR-1::GFP signal upon exposure to *E. faecium* in an *hlh-26*–dependent manner (Fig 6E and 6F).

To confirm that BAR-1 acted downstream of HLH-26 during *E. faecium*–mediated protection, we tested whether BAR-1 overexpression could rescue the reduced protection phenotype of *hlh-26*(*ok1453*) animals. Overexpressed BAR-1 under the regulation of the intestine-specific promoter *Pvha-6* in *hlh-26*(*ok1453*) animals rescued the decreased protection phenotype of *hlh-26*(*ok1453*) animals (Fig 6G–6J). We also found that HLH-26 overexpression in the intestine is sufficient to promote *bar-1* expression (S12 Fig). Together, these results showed that *hlh-26* was required for probiotic protection against *S. enterica* infection by regulating the canonical Wnt/BAR-1 pathway.

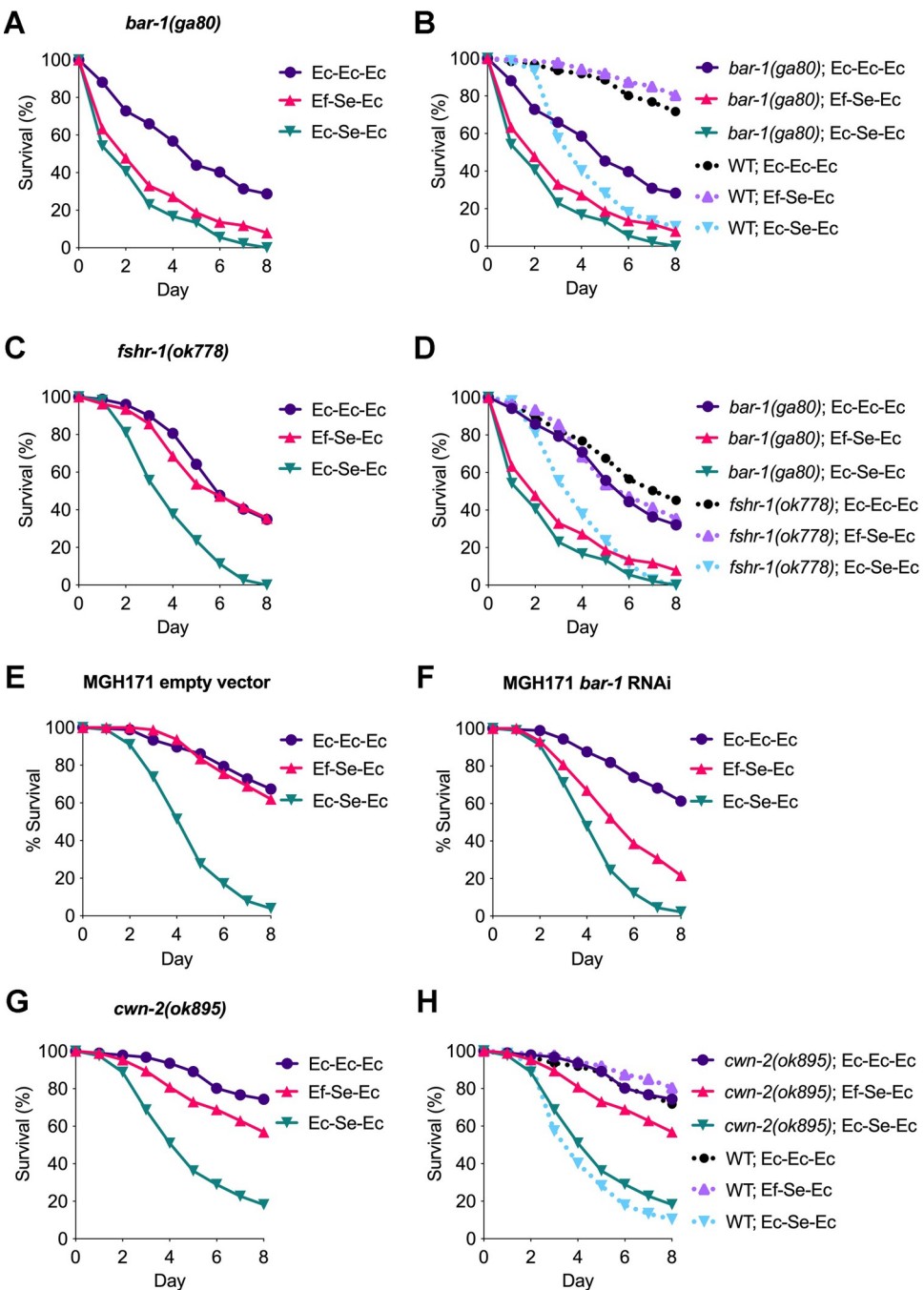

**Fig 5. Bar-1 is required in E. faecium protection against S. enterica infection.** (A) Survival curve assaying *E. faecium*–mediated protection in *bar-1(ga80)* animals. (B) Comparison of *E. faecium*–mediated protection between *bar-1(ga80)* animals and wild-type N2 animals. WT; Ec-Ec-Ec versus *bar-1(ga80)*; Ec-Ec-Ec, *P* < 0.0001; WT; Ef-Se-Ec versus *bar-1(ga80)*; Ef-Se-Ec, *P* < 0.0001; WT; Ec-Se-Ec versus *bar-1(ga80)*; Ec-Se-Ec, *P* < 0.0001. (C) Survival curve assaying *E. faecium*–mediated protection in *fshr-1(ok778)* animals. (B) Comparison of *E. faecium*–mediated protection between *bar-1(ga80)* animals and *fshr-1(ok778)* animals. *Fshr-1(ok778)*; Ec-Ec-Ec versus *bar-1(ga80)*; Ec-Ec-Ec, *P* < 0.05; *fshr-1(ok778)*; Ef-Se-Ec versus *bar-1(ga80)*; Ef-Se-Ec, *P* < 0.0001; *fshr-1(ok778)*; Ec-Se-Ec versus *bar-1(ga80)*; Ec-Se-Ec, *P* < 0.0001. Survival curves assaying *E. faecium*–mediated protection in the intestine-specific RNAi strain MGH171 feeding on HT115 expressing empty vector or (F) *bar-1* RNAi animals. (G) Survival curve assaying *E. faecium*–mediated protection in *cwn-2(ok895)* animals. (H) Comparison of *E. faecium*–mediated protection between *cwn-2(ok895)* animals and wild-type N2 animals. WT; Ec-Ec-Ec versus *cwn-2(ok895)*; Ec-Ec-Ec, *P* = NS; WT; Ef-Se-Ec versus *cwn-2(ok895)*; Ef-Se-Ec, *P* < 0.001; WT; Ec-Se-Ec versus *cwn-2(ok895)*; Ec-Se-Ec, *P* = NS. The data underlying all the graphs shown in the figure can be found in S1 Data. NS, nonsignificant; RNAi, RNAi interference; WT, wild-type.

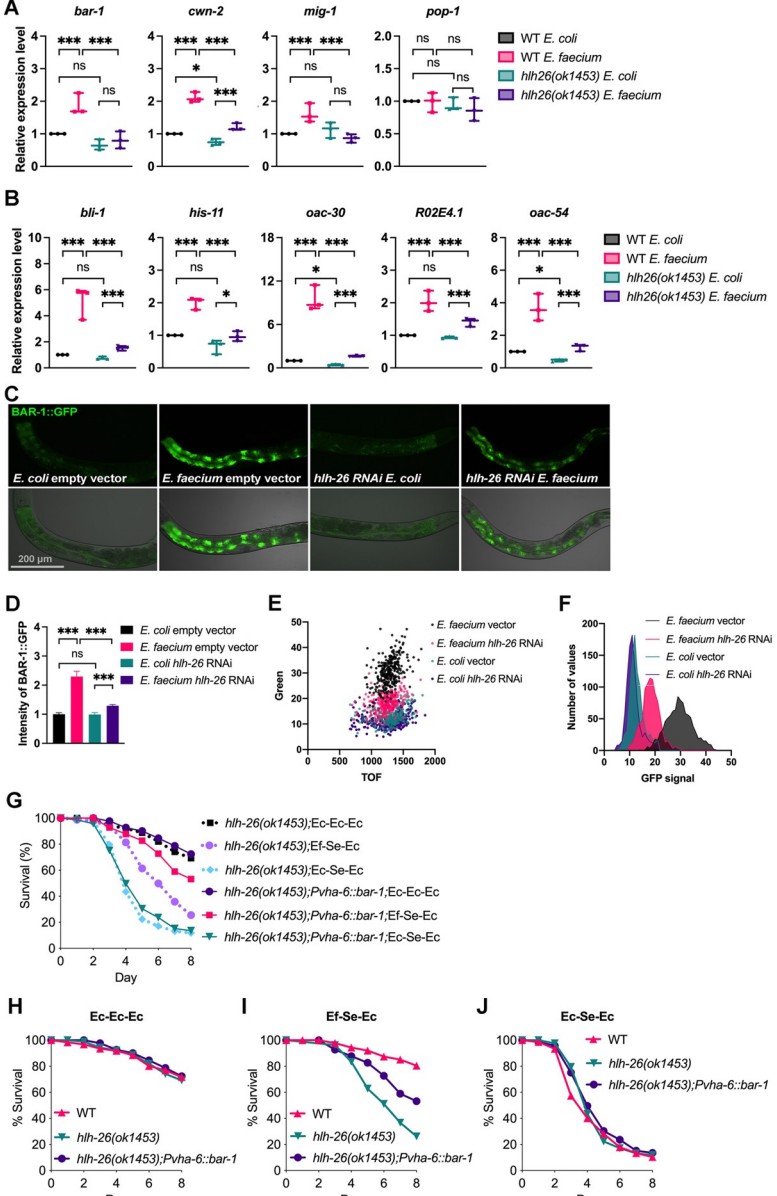

**Fig 6. HLH-26 is required for probiotic protection against *S. enterica* infection by regulating the canonical Wnt/ BAR-1 pathway.** (A) qRT-PCR validation of Wnt component genes (*N* = 3 biological replicates). Values are expressed as the fold difference compared with wild-type worms fed on *E. coli* ± SD by one-way ANOVA with Tukey's multiple comparisons test, *$P < 0.05$, **$P < 0.01$, ***$P < 0.001$. (B) qRT-PCR validation of overlapping genes from Wnt targets and *hlh-26*–down-regulated genes (*N* = 3 biological replicates). Values are expressed as the fold difference compared with wild-type worms fed on *E. coli* ± SD by one-way ANOVA with Tukey's multiple comparisons test, *$P < 0.05$, **$P < 0.01$, ***$P < 0.001$. (C) Fluorescence micrographs of control or *hlh-26* RNAi animals expressing GFP fused to BAR-1 (BAR-1::GFP) after a 24-h exposure to *E. faecium* or *E. coli*. (D) Quantification of the animals shown in (C), *N* = 3 biological replicates. *$P < 0.05$, **$P < 0.01$, ***$P < 0.001$. (E) Dot-plot representation of green fluorescence intensity versus TOF of vector or *hlh-26* RNAi animals exposed to *E. faecium* or *E. coli* (*n* = 300). (F) Frequency distribution of the green fluorescence of vector or *hlh-26* RNAi animals exposed to *E. faecium* or *E. coli*. Three independent experiments were performed. *E. coli* vector versus *E. coli hlh-26* RNAi, *P* = NS; *E. coli* vector versus *E. faecium* vector, *P* < 0.001; *E. faecium* vector versus *E. faecium hlh-26* RNAi, *P* < 0.001. (G) Comparation of *E. faecium*–mediated protection between *hlh-26(ok1453)* animals and *hlh-26(ok1453); Pvha-6::bar-1* rescue animals. *hlh-26(ok1453)*; Ec-Ec-Ec versus *hlh-26(ok1453); Pvha-6::bar-1*;Ec-Ec-Ec, *P* = NS; *hlh-26(ok1453)*; Ef-Se-Ec versus *hlh-26 (ok1453); Pvha-6::bar-1*; Ef-Se-Ec, *P* < 0.0001; *hlh-26(ok1453)*; Ec-Se-Ec versus *hlh-26(ok1453); Pvha-6::bar-1*; Ec-Se-Ec, *P* = NS. (H) WT, *hlh-26(ok1453)*, and *hlh-26(ok1453); Pvha-6::bar-1* animals were grown with Ec-Ec-Ec treatment and scored for survival. WT animals versus *hlh-26(ok1453)*, *P* = NS; *hlh-26(ok1453); Pvha-6::bar-1*, *P* = NS. (I) WT,

*hlh-26(ok1453)*, and *hlh-26(ok1453); Pvha-6::bar-1* animals were grown with Ef-Se-Ec treatment and scored for survival. WT animals versus *hlh-26(ok1453)*, $P$ = NS; *hlh-26(ok1453);Pvha-6::bar-1*, $P < 0.0001$. *hlh-26(ok1453)* animals versus *hlh-26(ok1453); Pvha-6::bar-1*, $P < 0.0001$. (J) WT, *hlh-26(ok1453)*, and *hlh-26(ok1453); Pvha-6::bar-1* animals were grown with Ec-Se-Ec treatment and scored for survival. WT animals versus *hlh-26(ok1453)*, $P$ = NS; *hlh-26(ok1453); Pvha-6::bar-1*, $P$ = NS. The data underlying all the graphs shown in the figure can be found in S1 Data. ns, nonsignificant; qRT-PCR, quantitative reverse transcription PCR; RNAi, RNAi interference; TOF, time of flight; WT, wild-type.

## Discussion

Probiotics exert beneficial effects on host health through different mechanisms of action, such as the production of antimicrobial substances, competition with pathogens, and immunomodulation [12]. However, the mechanisms by which probiotics modulate immune function to protect the host against pathogen infections in the intestine remain obscure. In this study, we uncovered a role for HLH-26 in the protection against infection caused by exposure to the probiotic *E. faecium*. We provided evidence indicating that HLH-26–dependent immune genes were activated upon exposure to *E. faecium* to defend against *S. enterica* infection. We also established that the canonical Wnt/BAR-1 pathway, which is transcriptionally regulated by HLH-26, was required in *E. faecium*–mediated protection against *S. enterica* infection (Fig 7).

Both HLH-26 and REF-1 were found to be required for *E. faecium*–mediated protection against *S. enterica* infection (Fig 1C and 1D). This finding is consistent with the CACGTG E-box binding capability of HLH-26 and REF-1 [28], and the presence of the CACGTG E-box in several genes up-regulated by *E. faecium* exposure (S2 Table). We also tested HLH-30, another REF-1 family member that can also bind to the CACGTG E-box and functions in the defense response [29]. However, HLH-30 was not required for *E. faecium*–mediated protection against *S. enterica* infection, potentially because different TFs exhibit different binding abilities to E-box and E-box-like sequences. Indeed, HLH-30 favors a flanking T, while HLH-26 favors an A or G and REF-1 disfavors a T, indicating that flanking nucleotides may play an important role in the functional TF divergence [28]. Other unidentified REF-1 family members may also act in regulating immune function by binding to the promoter of probiotic-responding genes. *C. elegans* has 934 annotated TFs [16], and we identified transcription factors by alignment of conserved binding motifs to the reported motifs of 292 *C. elegans* TFs. The discovery of a conserved binding motif for more TFs indicates that more TFs involved in regulating immune function in response to probiotics remain to be identified.

The Wnt signaling pathway is one of the most widely used and essential extracellular signaling mechanisms in metazoans, and components of this pathway are conserved from hydra to humans [22]. Besides regulating multiple processes that are crucial for embryogenesis and adult tissue homeostasis [30], Wnt signaling also functions in the immune response by orchestrating phagocytosis, antimicrobial defense, and inflammatory cytokine responses [31–33]. BAR-1 encodes a *C. elegans* homolog of β-catenin that is critical for conferring resistance to *S. aureus* [9]. Mutation of *bar-1* abrogates the induction of some *E. faecium*- and *E. faecalis*-activated genes [13]. Because BAR-1 plays a critical role in the response to *E. faecium* and HLH-26 controls the expression of differentially expressed Wnt genes upon *E. faecium* exposure, it is reasonable to postulate that both HLH-26 and BAR-1 function in the same pathway that activates immune genes in response to *E. faecium*. Treatment with *E. faecium* increased the protein level of β-catenin BAR-1, and knockdown of *hlh-26* resulted in a significant reduction of BAR-1 (Fig 6C–6F). In addition, the mRNA level of Wnt/BAR-1 and downstream targets increased upon exposure to *E. faecium* in an *hlh-26*–dependent manner (Fig 6B). In summary, we identified a new mechanism by which probiotics might modulate immune function and regulate defense against pathogen infection. We found that HLH-26 was required for

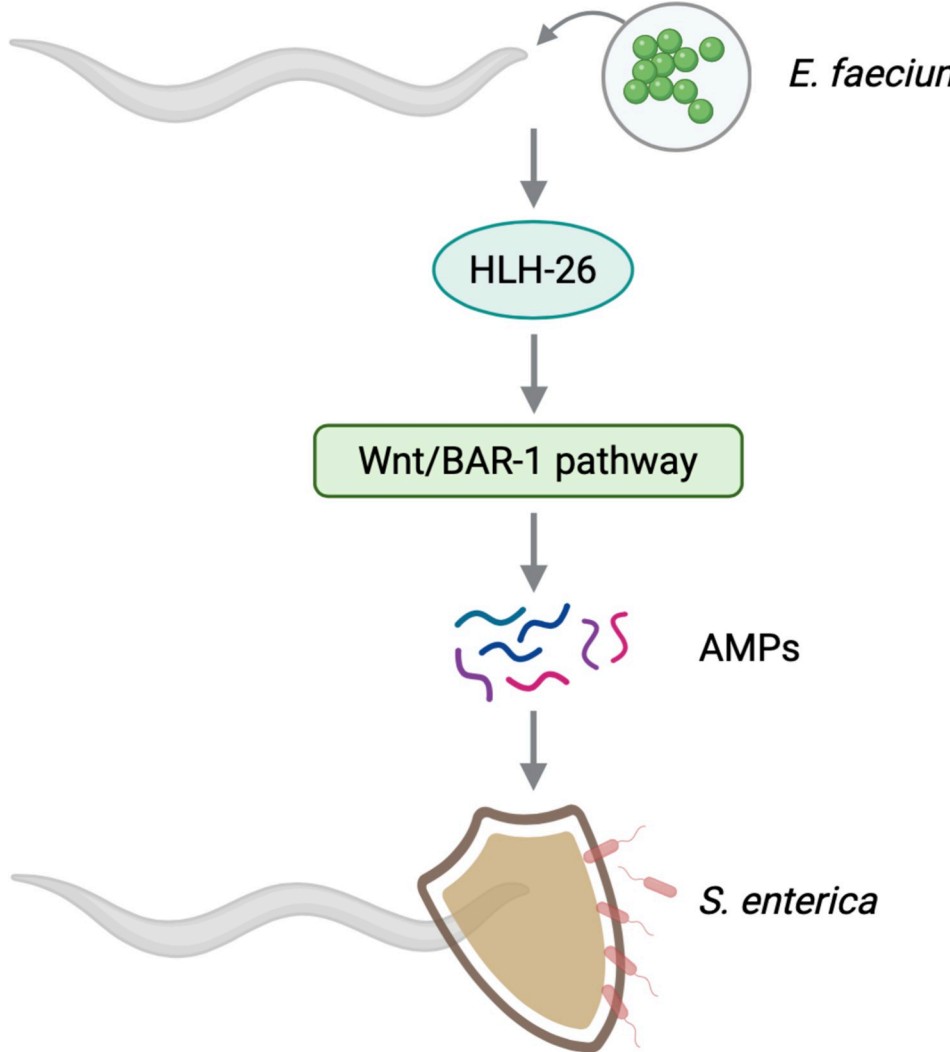

**Fig 7. Transcription factor HLH-26 mediates probiotic protection via Wnt/BAR-1 Signaling.** HLH-26 protection against *S. enterica* infection is partly due to activation of the canonical Wnt/BAR-1 pathway. The binding of Wnt to its receptors leads to β-catenin stabilization and translocation to the nucleus, coactivating target gene transcription. Expression of Wnt/BAR-1 target genes enhances antimicrobial peptide production, which inhibits pathogen infection. AMPs, antimicrobial peptides.

protection against *S. enterica* infection induced by exposure to the probiotic *E. faecium*. We further established that the canonical Wnt/BAR-1 pathway, which was regulated by HLH-26, was also required in *E. faecium*–mediated protection against *S. enterica* infection. Our results highlight a critical role for HLH-26/Wnt/BAR-1 in the control of probiotic-mediated immune activation and the antimicrobial response.

## Materials and methods

### Bacterial strains

The following bacterial strains were used: *E coli* OP50, *E. coli* HT115(DE3), *E. faecium* NCTC 7171, *S. enterica* serovar Typhimurium 14028s. *E. faecium* was grown in brain–heart infusion (BHI) medium at 37˚C, and the cultures of other bacteria were grown in Luria–Bertani (LB) broth at 37˚C.

## *C. elegans* strains and growth conditions

All *C. elegans* strains were maintained on nematode growth medium (NGM) and fed *E. coli* strain OP50. The *C. elegans* strains wild-type N2 Bristol, RB1337 *hlh-26(ok1453)*, PS3931 *ref-1 (ok288)*, VC1689 *ces-2(gk1020)*, RB824 *ztf-11(ok646)*, RB772 *atf-6(ok551)*, RB911 *fshr-1 (ok778)*, EW15 *bar-1(ga80)*, *WM300 (alg-4(ok1041) III; alg-3(tm1155) IV)*, NL3307 *rsd-2 (pk3307)*, MT1524 *lin-28(n719)*, SX922 *prg-1(n4357)*, VC3044 *dbl-1(ok3749)*, and JIN1375 *hlh-30(tm1978)* used in the treatment–infection assays were obtained from the *Caenorhabditis* Genetics Center (University of Minnesota, Minneapolis, MN). The germline-specific RNAi strain DCL569 (*mkcSi13 II;rde-1(mkc36) V*), neuron-specific RNAi strain TU3401 (*sid-1 (pk3321) V;uIs69 V*), and gut-specific RNAi strain MGH171 (*sid-1(qt9) V;alxIs9*) were obtained from the *Caenorhabditis* Genetics Center. EW25 (*unc-30(e191)IV; gaIs45 [pDE218 (bar-1::bar-1::gfp); unc-30+]*) was a gift from David Eisenmann [27]. Detailed strain information is listed in S9 Table.

## Treatment infection assay

Synchronized young adult worms were washed 3 times in M9 buffer and transferred to *E. faecium* lawns grown on 2% agar BHI plates for colonization for 1 day at 25˚C. The worms were then washed and transferred to *S. enterica* lawns grown on slow-killing (SK) plates (modified NGM agar plates (0.35% instead of 0.25% peptone)) for infection for another day at 25˚C. Finally, the worms were washed and transferred to lawns of OP50 grown on NGM plates (Day 1). They were maintained on OP50-NGM plates at 25˚C, and survival was scored. *E. coli* OP50 grown on NGM plates were used as a control for *E. faecium* treatment or *S. enterica* infection. For protection and infection assays using transgenic strains with HLH-26 or BAR-1 intestinal specific rescue, non-roller animals with GFP signals in the intestine were picked after synchronization.

## Continuous infection assay

Synchronized young adult worms were washed and transferred to *E. faecium* lawns grown on 2% agar BHI plates for 1 day of colonization. Then, the worms were washed and transferred to lawns of *S. enterica* grown on SK plates (Day 0). Worms were maintained on *S. enterica*-SK plates at 25˚C, and survival was scored.

## CFU measurements

To quantify intestinal bacterial loads, 6 animals were rinsed using drops of S buffer and allowed to crawl free of bacteria on a sterile plate. These animals were transferred into 100 μL of PBS plus 0.01% Triton X-100 and ground individually. Serial dilutions of worm homogenate were seeded on *Salmonella-Shigella* agar (BD BBL) plates, and plates were incubated at 37˚C overnight. Single colonies were counted the next day and represented as the number of bacterial cells or CFU per animal.

## Generation of transgenic *C. elegans*

The *hlh-26* DNA was amplified from genomic DNA of Bristol N2 *C. elegans* adult worms as template using the primers presented in S10 Table. The *hlh-26* promoter sequence (2,363 bp) was amplified with primers *hlh-26*_-2.3k_*SaL*I F and *hlh-26*_+1 *BamH*I R (S10 Table). The amplified *hlh-26* promoter sequence was cloned into plasmid pPD95.75_GFP between the *Sal* I and *BamH* I sites to generate the clone pPD95.75_*hlh-26p_gfp*. Young adult hermaphrodite wild-type animals were transformed by microinjection of the plasmids into the gonads. A

mixture containing the pPD95.75_*hlh-26p*_GFP plasmids (25 ng/μL) and *coel::rfp* (25 ng/μL) as a transformation marker was injected into the animal. Successful transformation was determined by identification of the selection marker as red dots. For HLH-26 gut-specific rescue, *hlh-26* encoding sequence was amplified with primers *hlh-26*_ATG_*Sal*I F and *hlh-26*_TAA_*Sma*I R (S10 Table). The amplified *hlh-26* DNA was cloned under the *vha-6* promoter in plasmid pPD95.77_*Pvha-6*_SL2 between the *Sal* I and *Sma* I sites to generate the expression clone pPD95.77_*Pvha-6_hlh-26*_SL2. The construct was purified and sequenced. Young adult hermaphrodite *hlh-26(ok1453)* animals were transformed by microinjection of plasmids into the gonads. A mixture containing the pPD95.77_*Pvha-6_hlh-26*_SL2 plasmids (25 ng/μL) and pRF4_ *rol-6(su1006)* (25 ng/μL) as a transformation marker was injected into the worm. Successful transformation was determined by identification of the selection marker as a dominant roller. pPD95.77_*Pvha-6_bar-1*_SL2 was constructed and injected in the same manner. *bar-1* encoding sequence was amplified with primers *bar-1*_ATG_*Sal*I F and *bar-1*_TAA_*Sma*I R (S10 Table). At least 3 independent lines carrying extra chromosomal arrays were obtained for each construct.

## SagA treatment

Briefly, log phage *E. coli* BL21 (DE3) cultures with empty vector or expressing SagA grown in LB broth containing ampicillin (100 μg/ml) at 37°C overnight were harvested, 1:10 concentrated, and plated onto NGM plates containing 100 mg/ml ampicillin and 3 mM isopropyl-β-D-thiogalactoside (IPTG) (RNAi plates). SagA-expressing bacteria were allowed to grow and express SagA overnight at 15°C. Synchronized young adults were transferred to SagA-expressing bacterial lawns and incubated 1 day at 25°C. The worms were then washed and transferred to *S. enterica* lawns grown on SK plates (modified NGM agar plates (0.35% instead of 0.25% peptone)) for infection for another day at 25°C. Finally, the animals were washed and transferred to lawns of OP50 grown on NGM plates (Day 1). They were maintained on OP50-NGM plates at 25°C, and survival was scored. *E. coli* OP50 grown on NGM plates was also used as a control for SagA treatment or *S. enterica* infection.

## RNAi and *E. faecium* treatment

RNAi was used to generate loss-of-function RNAi phenotypes by feeding nematodes the *E. coli* strain HT115 (DE3) expressing double-stranded RNA (dsRNA) homologous to a target gene [34]. Briefly, *E. coli* HT115 (DE3) with appropriate vectors was grown in LB broth containing ampicillin (100 μg/ml) and tetracycline (12.5 μg/ml) at 37°C overnight and plated onto NGM plates containing 100 mg/ml ampicillin and 3 mM IPTG (RNAi plates). RNAi-expressing bacteria were allowed to grow overnight at 37°C. Gravid adults were transferred to RNAi-expressing bacterial lawns and allowed to lay eggs for 2 h. The gravid adults were removed, the eggs were allowed to develop at 20°C into young adults, and then exposed to *E. faecium*. RNAi targeting *unc-22* was included as a positive control to account for the RNAi efficiency. The RNAi clone was from the Ahringer RNAi library (Open BioSource), and it was sequenced using universal GeneWiz M13 primers.

## RNA sequencing and analyses

Approximately 100 gravid animals were placed on 10-cm NGM plates (seeded with *E. coli* OP50) for 3 h to obtain a synchronized population, which developed and grew to the L4 larval stage at 20°C. Synchronized L4 worms were washed 3 times in M9 buffer and transferred to *E. faecium* lawns grown on 2% agar BHI plates for colonization for 8 h at 25°C [13]. Animals were washed off the plates with M9, frozen in QIAzol using ethanol/dry ice, and stored at

−80˚C prior to RNA extraction. Total RNA was extracted using the RNeasy Plus Universal Kit (Qiagen, the Netherlands). Residual genomic DNA was removed using TURBO DNase (Life Technologies, Carlsbad, CA). A total of 6 μg of total RNA was reverse-transcribed with random primers using the High-Capacity cDNA Reverse Transcription Kit (Applied Biosystems, Foster City, CA).

Library construction and RNA sequencing on the BGISEQ-500 platform was performed according to [35,36], and paired-end reads of 100 bp were obtained for subsequent data analysis. The RNA sequence data were analyzed using a workflow constructed for Galaxy (https://usegalaxy.org) [37–39]. The RNA reads were aligned to the *C. elegans* genome (WS230) using the aligner STAR. Counts were normalized for sequencing depth and RNA composition across all samples. Differential gene expression analysis was then performed using normalized samples. Genes exhibiting at least a 2-fold change were considered differentially expressed. The differentially expressed genes were subjected to SimpleMine tools from WormBase (https://www.wormbase.org/tools/mine/simplemine.cgi) to obtain information such as the WormBase ID and gene names, which were employed for further analyses. GO analysis was performed using the WormBase IDs in the DAVID Bioinformatics Database (https://david.ncifcrf.gov) [40] and a *C. elegans* data enrichment analysis tool (https://wormbase.org/tools/enrichment/tea/tea.cgi).

### RNA isolation and quantitative reverse transcription PCR (qRT-PCR)

Animals were synchronized, and total RNA extraction was performed following the above-described protocol. qRT-PCR was conducted using the Applied Biosystems One-Step Realtime PCR protocol with SYBR Green fluorescence (Applied Biosystems) on an Applied Biosystems 7900HT real-time PCR machine in 96-well plate format. The reactions were analyzed as outlined by the manufacturer (Applied Biosystems). The relative fold changes of the transcripts were calculated using the comparative CT($2^{-\Delta\Delta CT}$) method and normalized to pan-actin (*act-1, -3, -4*). The cycle thresholds of the amplification were determined using StepOnePlus Real-Time PCR System Software v2.3 (Applied Biosystems). All samples were run in triplicate. The primer sequences are presented in S10 Table.

### Fluorescence imaging

Fluorescence imaging was conducted according to [41,42] with slight modifications. Briefly, animals were anesthetized using an M9 salt solution containing 50 mM sodium azide and mounted onto 2% agar pads. The animals were then visualized using a Leica M165 FC fluorescence stereomicroscope. Fluorescence was quantified using Fiji-ImageJ (https://imagej.net/Fiji/Downloads).

### COPAS biosorter GFP analysis

Expression levels of the transgenic reporter BAR-1::GFP were analyzed using the COPAS Biosort instrument (Union Biometrica). Synchronized young adult EW25 (BAR-1::GFP) animals on *hlh-26* RNAi or control plates were exposed to *E. faecium* or *E. coli* for 24 h and washed in M9 buffer before analysis.

### Statistical analysis

The two-tailed Student *t* test for independent samples was used to analyze the data. To compare the means of more than 2 groups, one-way ANOVA with a post hoc analysis was performed. All experiments were repeated at least 3 times, and error bars represent the standard

deviation, unless otherwise indicated. The data were judged to be statistically significant when $P < 0.05$. "n" represents the number of animals in each experiment. "NS" indicates nonsignificant, and the asterisk "*" indicates a significant difference; *$P < 0.05$, **$P < 0.01$, ***$P < 0.001$. The Kaplan–Meier method was used to calculate the survival fractions, and the statistical significance between survival curves was determined using the log-rank test. All experiments were performed at least 3 times. Raw data and statistical data analysis were listed in S1 Data.

## Supporting information

**S1 Fig. Protection effect from both live and heat-killed *E. faecium* against *S. enterica* pathogenesis.** Survival curve from a treatment–1-day infection assay showing that both live and heat-killed *E. faecium* inhibits *S. enterica* pathogenesis. Animals were treated on live or heat-killed *E. faecium* for 1 day before infection with *S. enterica* for 1 day. Animals were then transferred onto *E. coli* OP50 plates for the rest of the assay, and survival was scored. The data underlying all the graphs shown in the figure can be found in S1 Data.
(TIF)

**S2 Fig. *E. faecium* protection in *hlh-26(ok1453);ref-1(ok288)* mutant.** (A) Survival curve assaying *E. faecium*–mediated protection of *hlh-26(ok1453);ref-1(ok288)* double mutant. Survival curves are representative assays of 3 independent experiments. *n* = 60 to 90. (B) Survival curves of *ref-1(ok288)*, *hlh-26(ok1453)*, and *hlh-26(ok1453);ref-1(ok288)* animals grown on *E. faecium* for 1 day, infected with *S. enterica* for 1 day, and transferred to *E. coli*. *ref-1(ok288)* animals versus *hlh-26(ok1453)*, $P < 0.0001$; *ref-1(ok288)* animals versus *hlh-26(ok1453);ref-1(ok288)* animals, $P < 0.0001$. *hlh-26(ok1453)* animals versus *hlh-26(ok1453);ref-1(ok288)* animals, $P$ = NS. The data underlying all the graphs shown in the figure can be found in S1 Data. Ec, *E. coli* OP50; Ef, *E. faecium*; NS, nonsignificant; Se, *S. enterica*.
(TIF)

**S3 Fig. Heat-killed *E. coli* does not protect against *S. enterica* infection.** (A) Schematic of the *E. faecium* treatment and 1-day infection assay. Animals were grown on *E. faecium* lawns for 1 day before infection with *S. enterica* for 1 day. Animals were then transferred onto *E. coli* OP50 plates for the remainder of the assay, and survival was scored. Control animals were fed heat-killed *E. coli* OP50 throughout the assay. Survival curves assaying *E. faecium*–mediated protection in (B) wild-type and (C) *hlh-26(ok1453)* animals. The data underlying all the graphs shown in the figure can be found in S1 Data. Ec, *E. coli* OP50; Ef, *E. faecium*; Se, *S. enterica*; WT, wild-type.
(TIF)

**S4 Fig. *hlh-26* is required for SagA-mediated protection.** (A) Schematic of the *E. faecium* treatment and 1-day infection assay. Animals were pretreated with *E. coli* BL21 harboring empty vector or expressing SagA followed by *S. enterica* infection. Animals were then transferred onto *E. coli* OP50 plates for the remainder of the assay, and survival was scored. Survival curves assaying SagA-mediated protection in (B) wild-type and (C) *hlh-26(ok1453)* animals. (D) Fluorescence micrographs of animals expressing GFP fused to HLH-26 (HLH-26::GFP) after 24-h fed on *E. coli* BL21 harboring empty vector or expressing SagA. (E) Quantification of the animals shown in (D), *N* = 3 biological replicates. ***$P < 0.001$. The data underlying all the graphs shown in the figure can be found in S1 Data. Ec, *E. coli* OP50; EcSagA, *E. coli* BL21 expressing SagA; EcVector, *E. coli* BL21 harboring empty vector; Se, *S. enterica*.
(TIF)

**S5 Fig. Survival curve of *C. elegans* on *E. faecium* or *S. enterica*.** (A) WT and *hlh-26(ok1453)* animals were grown on *E. faecium* and scored for survival. WT animals versus *hlh-26(ok1453)*, *P* = NS. (B) WT and *hlh-26(ok1453)* animals were grown on *S. enterica* and scored for survival. WT animals versus *hlh-26(ok1453)*, *P* = NS. The data underlying all the graphs shown in the figure can be found in S1 Data. NS, nonsignificant; WT, wild-type. (TIF)

**S6 Fig. mRNA levels of coexpressed genes.** qRT-PCR validation of selected coexpressed genes from which the upstream regulator *hlh-26* was predicted. Values are expressed as the fold difference compared with wild-type worms fed on heat-killed *E. coli* ± SD by one-way ANOVA with Tukey's multiple comparisons test, *$*P < 0.05$, $**P < 0.01$, $***P < 0.001$. The data underlying all the graphs shown in the figure can be found in S1 Data. ns, nonsignificant; qRT-PCR, quantitative reverse transcription PCR; WT, wild-type. (TIF)

**S7 Fig. *E. faecium* protection in *C. elegans* with *hlh-26* tissue-specific RNAi.** Survival curves assaying *E. faecium*–mediated protection in the germline-specific RNAi strain DCL569 fed on HT115 expressing (A) empty vector or (B) *hlh-26* RNAi animals. Survival curves assaying *E. faecium*–mediated protection in the neuron-specific RNAi strain Tu3401 fed on HT115 expressing (A) empty vector or (B) *hlh-26* RNAi animals. The data underlying all the graphs shown in the figure can be found in S1 Data. RNAi, RNAi interference. (TIF)

**S8 Fig. Enrichment analysis of HLH-26–regulated genes.** (A) Enrichment analysis of up-regulated genes. (B) Enrichment analysis of down-regulated genes. The data underlying all the graphs shown in the figure can be found in S1 Data. (TIF)

**S9 Fig. *E. faecium* protection in *C. elegans* mutants.** Survival curve assaying *E. faecium*–mediated protection in (A) *alg-4(ok1041); alg-3(tm1155)*, (B) *rsd-2(pk3307)*, (C) *lin-28(n719)*, (D) *prg-1(n4357)*, (E) *dbl-1(ok3749)*, and (F) *hlh-30(tm1978)*. Survival curves are representative assays of 3 independent experiments. *n* = 60 to 90. The data underlying all the graphs shown in the figure can be found in S1 Data. Ec, *E. coli* OP50; Ef, *E. faecium*; Se, *S. enterica*. (TIF)

**S10 Fig. Survival curve on *S. enterica*.** WT, *bar-1(ga80)*, and *fshr-1(ok778)* animals were grown on modified NGM agar (0.35% peptone instead of 0.25% peptone) for infection with *S. enterica* and scored for survival. WT animals versus *bar-1(ga80)*, *P* < 0.0001; *fshr-1(ok778)*, *P* < 0.0001. *fshr-1(ok778)* animals versus *bar-1(ga80)*, *P* < 0.0001. The data underlying all the graphs shown in the figure can be found in S1 Data. NGM, nematode growth medium; WT, wild-type. (TIF)

**S11 Fig. Survival curve of *cwn-2(ok895)* animals on *S. enterica*.** WT and *cwn-2(ok895)* animals were grown on *S. enterica* and scored for survival. WT animals versus *cwn-2(ok895)*, *P* = NS. The data underlying all the graphs shown in the figure can be found in S1 Data. NS, nonsignificant; WT, wild-type. (TIF)

**S12 Fig. mRNA levels of *bar-1* in *hlh-26(ok1453)* animals and *hlh-26(ok1453)* animals expressing HLH-26 in the intestine.** mRNA levels of *bar-1* in wild-type, *hlh-26(ok1453)* animals, and *hlh-26(ok1453)* animals overexpressing HLH-26 fed on *E. coli* or exposed to *E.*

*faecium* were detected using qRT-PCR. Values are expressed as the fold difference compared with wild-type animals fed on *E. coli* ± SD by one-way ANOVA with Tukey's multiple comparisons test, $^{**}P < 0.01$, $^{***}P < 0.001$. The data underlying all the graphs shown in the figure can be found in S1 Data. qRT-PCR, quantitative reverse transcription PCR; WT, wild-type. (TIF)

**S1 Data. Raw data and statistical data analysis.**
(XLSX)

**S1 Table. *E. faecium*–regulated genes that are coexpressed.**
(XLSX)

**S2 Table. Common binding motifs for the 30 top ranked coexpressed groups.**
(XLSX)

**S3 Table. Genes potentially regulated by top 5 TFs.**
(XLSX)

**S4 Table. Sequence.**
(XLSX)

**S5 Table. Term.**
(XLSX)

**S6 Table. Down-regulated immune genes in hlh-26(ok1453) compared with wild-type animals exposed to *E. faecium*.**
(XLSX)

**S7 Table. Gene enrichment analysis of HLH-26–regulated genes.**
(XLSX)

**S8 Table. Representation factors for down-regulated genes in hlh-26(ok1453) animals ($n = 1,172$).**
(XLSX)

**S9 Table. Strains.**
(XLSX)

**S10 Table. Primers.**
(XLSX)

## Acknowledgments

We thank David M. Eisenmann (University of Maryland, Baltimore, MD 21250) for providing BAR-1::GFP strain, Howard Hang (Scripps, La Jolla, CA 92037) for *pET21a-sagA* plasmid, and Michael McClelland (University of California, Irvine, CA 92697) for providing the *Salmonella enterica* serovar Typhimurium 14028s strain.

## Author Contributions

**Conceptualization:** Yu Sang, Alejandro Aballay.

**Data curation:** Yu Sang.

**Formal analysis:** Yu Sang, Alejandro Aballay.

**Funding acquisition:** Alejandro Aballay.

**Investigation:** Yu Sang, Jie Ren.

**Methodology:** Yu Sang, Jie Ren.

**Project administration:** Alejandro Aballay.

**Supervision:** Alejandro Aballay.

**Writing – original draft:** Yu Sang.

**Writing – review & editing:** Alejandro Aballay.

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
