## [Editor Report · Decision Letter 0]

5 Aug 2021

Dear Dr. Aballay, 

Thank you for submitting your manuscript entitled "The REF-1 family transcription factor HLH-26 controls the Wnt/BAR-1 in probiotic-mediated protection against gut infection" for consideration as a Research Article by PLOS Biology. 

Your manuscript has now been evaluated by the PLOS Biology editorial staff and I am writing to let you know that we would like to send your submission out for external peer review.

Please re-submit your manuscript within two working days, i.e. by Aug 07 2021 11:59PM.

Kind regards,

Ines Alvarez-Garcia

Senior Editor

PLOS Biology

---

## [Decision Letter · Decision Letter 1]

23 Sep 2021

Dear Alejandro,

Thank you for submitting your manuscript entitled "The REF-1 family transcription factor HLH-26 controls the Wnt/BAR-1 in probiotic-mediated protection against gut infection" for consideration as a Research Article at PLOS Biology. Thank you also for your patience as we completed our editorial process, and please accept my apologies for the delay in providing you with our decision. Your manuscript has been evaluated by the PLOS Biology editors, an Academic Editor with relevant expertise, and by two independent reviewers.

You will see that the reviewers appreciate the interest of the conclusions, but also raise several concerns that would need to be addressed in order for us to consider the manuscript further for publication. Reviewer 1 asks for a better description of how the candidate transcription factors were selected, clarification of several points and suggests some refocusing of the manuscript. Reviewer 2 thinks you need to show if HLH-30 is activated in response to the proteolytic activity of SagA and also to perform an experiment to explore the relevance of the innate immune genes for the protection against S. enterica, among other issues.

In light of the reviews (attached below), we will not be able to accept the current version of the manuscript, but we would welcome re-submission of a revised version that takes into account the reviewers' comments. We cannot make any decision about publication until we have seen the revised manuscript and your response to the reviewers' comments. Your revised manuscript is also likely to be sent for further evaluation by the reviewers.

We expect to receive your revised manuscript within 3 months. 

**IMPORTANT - SUBMITTING YOUR REVISION**

3. Resubmission Checklist

a) *Published Peer Review*

b) *PLOS Data Policy*

Sincerely,

Ines

--

Ines Alvarez-Garcia, PhD

Senior Editor

PLOS Biology

Reviewers’ comments

Rev. 1:

Summary

In this study, the authors demonstrated that pretreatment with the probiotic species E. faecium protects C. elegans against S. enterica infection. Through a meta-analysis on the gene expression dataset and transcription factors (TFs), the authors identified 28 TFs that are potentially contribute to such protection. The authors then tested the top 5 candidate TFs and found that the E. faecium mediated protection is dependent on the REF-1 family transcription factor HLH-26. The authors next characterized that HLH-26 is expressed in the intestine and the E. faecium mediated protection is dependent on the intestinal expression of HLH-26. To further describe the HLH-26-dependent E. faecium mediated protection, the authors conducted an RNAseq analysis and identified 3261 differentiated expressed genes by comparing the transcription profiles between hlh-26 mutant worms and wt worms grown on E. faecium. The authors found that many hlh-26-regulated genes were also controlled by many other pathways. To identify which pathways regulates the E. faecium mediated protection, the authors tested the key factors (mutant) in the candidate pathways on their functions in E. faecium mediated protection, and found only bar-1(β-catenin in Wnt signaling) is critical in the E. faecium mediated protection. Next, the authors confirmed that the expression of not only bar-1 but also other Wnt signaling components (cwn-2 and mlg-1), as well as downstream effectors, are all hlh-26 dependent. Finally, the authors validated that the function of bar-1 in E. faecium mediated protection is also hlh-26 dependent. Together, the authors propose a function for the uncharacterized REF-1-family transcription factor HLH-26 in probiotic E. faecium mediated protection against pathogenic S. enterica infection in C. elegans, and that Wnt signaling is a critical component of the HLH-26 immunity program.

I think this manuscript would be suitable for publication PLOS Biology after addressing the several concerns listed below:

Major points

1. It is unclear how the initial analysis was done to get the final number of 28 candidate TFs.

The identification of the initial 399 genes needs to be described. In S1 table, the 399 genes are from a previous publication (Yuen and Ausubel, 2018), but this literature was not cited in the text, only cited within the supplemental table 1.

Within these 399 genes, 244 genes were upregulated, and 155 genes were downregulated (Yuen and Ausubel 2018). It is unclear whether the authors did the co-expression category analysis differently on the up or down regulated genes. It is also unclear if the 30 (out of 125) top ranked co-expressed groups the authors picked to analyze include both up and down regulated genes or just up or down regulated genes. Moreover, the authors may need to specify what algorithm/criteria was used to rank the 125 groups and why the criteria were chosen.

For the motif analysis, it is unclear how many of the 158 motifs are for each of the 30 co-expressed groups, and how many of the motifs are shared between different groups.

Finally, it is unclear how many (and which) of the original 399 differentially expressed genes are (or predicted to be) regulated by the 28 TFs, especially the 5 top TFs tested, and how many of them are up or down regulated by these TFs.

Include full list of genes upregulated and downregulated upon exposure to E faecium in WT and hlh-26 mutant C. elegans with their respective p values. This means that 4 tables need to be added to Suppl information.

As you present these data, also discuss the immunity genes that are downregulated in C elegans exposed to E faecium.

2. According to the figure caption, Fig 1 B-G are one representative dataset of 3 independent experiments, please provide the survival data of the other 2 independent experiments as a supplemental file, include the statistical analysis of these survival assays (harzard ratio, median survival rate, p-values between conditions, etc.) and how many animals in each survival test. Please provide the results of these for all survival tests throughout the manuscript as well. A detailed experimental procedures and analysis of the survival assays is also missing in the material and method section, the author should provide it since the majority of the data is based on this assay. I would also suggest the author to provide the raw data of these experiments in a separated file.

3. Fig 2 A-C seems relatively not as important to be shown as main figure panels, because it simply addresses that the heat killed E. coli has no protection to Se infection, just like live E. coli in Fig 1. I would suggest moving it to a supplemental figure. However, on the other hand, It is very interesting to show if heat killed Efm protect against Se infection or not, and if the protection is from the Efm secreted metabolites or Efm intracellular materials.

Also, explicitly describe why in the transcriptomics analyses the authors use heat-killed E coli instead of heat killed E faecium as control. Later discuss heat-stable versus active metabolism as the E. faecium mediators of the probiotic response in C elegans.

4. It would be interesting to know how many of the 3261 differentially regulated genes share conserved DNA binding motif with HLH-26 to see how many of the genes are likely directly regulated by HLH-26, and what these genes are, and if they are up or down regulated. This information would help to demonstrate if the HLH-26 act as an activator or a repressor, and whether upregulation of immunity is a truly enriched component of the response. Also, it would be good to know how many of these 3261 genes overlap with the previous 399 genes, and how many of them overlap with the 70 genes that are predicted to be regulated by HLH-26. This analysis would narrow down the number of the genes and focus the study on HLH-26 directly regulated genes. Furthermore, the author should check the GO cluster of the overlapping genes and the predicted direct HLH-26 regulated genes (specially to check if the immune cluster also show up as highly enriched cluster).

For the differentially expressed genes that are predicted to be directly regulated by HLH-26 and overlap with the previous 70 genes, the authors may want to verify how many of them are HLH-26 dependent by qRT-PCR on the 4 conditions and to test if they functionally contribute to Efm protection against Se infection through RNAi followed by survival tests (if the genes are up-regulated genes). The result of these test are likely to provide an actual mechanism of probiotic-mediated immunity against S. enterica, the lack of which is currently my major concern.

5. Because the analysis on "hlh-26-regulated" genes are not narrowed down to the hlh-26 directly regulated genes as suggested in the last comment, it is not surprise that several other pathways also seem to control the "hlh-26-regulated" genes, and it is not surprise that most of those pathways show no role in the Efm protection (S5 Fig). I would suggest the authors analyze the validated hlh-26 dependent and "directly" regulated genes that contribute to the Efm protection (by RNAi as suggested in last comment), and to check the enriched pathways and the regulation of those genes.

6. The authors reasoned the survival of bar-1 animals decreasing on E. coli lawns as "susceptible to bacterial infections in general". The authors need to rephrase this since the reduction in survival could be due to Wnt/BAR-1 being required for C. elegans healthspan in general, and not specifically to susceptibility to bacterial infection.

7. How is E. faecium treatment performed in combination with RNAi? The authors controlled for heat-killed E coli not acting as a probiotic; however, does live E coli boost S. enterica resistance?

Please test and present these data.

8. In Fig 5 C-E, the authors try to make comparisons between bar-1 and fshr-1 to address the point that even the bar-1 and fshr-1 both reduce the survival to the same extent without Se infection, bar-1 contribute to the Efm protection against Se infection. The authors would want to include formal statistical analyses to support this claim (instead of a graphical representation).

9. Why do the authors modify the conditions to compare bar-1 to fshr-1-deficient worms? Please perform and present fshr-1 experiments in the same conditions than Fig. 1A and 1E. Is BAR-1::GFP a rescue construct? Is BAR-1 overexpression sufficient to activate the immune genes upregulated in Fig. 4B? Is BAR-1 predicted to bind to the promoters of those genes?

10. In Fig 3A-C, the authors implicate the intestinal function in HLH-26 mediated Efm protection. It is valid to test if the intestinal specific RNAi (MGH171 line) against bar-1 would reduce the survival of Efm treated worms. This condition may also reduce the general detrimental effects of bar-1 since the KD of bar-1 is restricted to intestine.

11. Present higher-resolution images of hlh-26P::GFP. Current Fig. 3A is confusing because it shows punctate GFP signal, instead of soluble (and hence homogeneous) cytoplasmic expression. As the authors are aware, soluble GFP generated from a promoter fusion is not expected to be trafficked to organelles.

12. Figure 3D shows rescue but not enhancement of pathogen resistance in animals overexpressing HLH-26. Discuss which would be the bottleneck to enhanced pathogen resistance in this context. Also, is HLH-26 overexpression in the intestine sufficient to promote bar-1 expression (perform qRTPCR of bar-1 in this line)

13. Could rollers ingest less S. enterica? Please perform survival experiment presented in Fig 1A but comparing WT N2 worms to N2 animals carrying pRF4 rol-6(su1006) alone.

14. The authors may need to present images of vha-6:::HLH-26::GFP worms (without SL2) to define whether HLH-26 is enriched in the intestinal nuclei of C elegans exposed to E faecium alone, S enterica alone, or E faecium followed by S enterica exposure.

15. Describe which is the N (independent biological replicates) in Figures 4C & D. Additionally, please check all figure legends to make sure N/n are described in a consistent manner. For example, n=number of animals and N=number of biological replicates.

16. The following expression may be inaccurate: "Downregulated genes in hlh-26(ok1453) animals included immune response genes". The data presented in Table S6 show that the analyzed genes were less expressed in hlh-26 than in WT worms, but we do not know if they were further downregulated or not as induced in the hlh-26 KO. In addition to the fold change, the authors may want to present the raw data for all 4 conditions (wild-type animals exposed to E. coli, wild-type animals exposed to E. faecium, hlh-26 animals exposed to E. coli, and hlh-26 animals exposed to E. faecium) for all genes listed in Table S6.

Similarly, "…that overlapped with genes that were downregulated in hlh-26 animals (S8 Table)." The genes referred to here may not actually be downregulated; instead, they may be less upregulated. Raw data are needed all around for all datasetsto avoid these confusions.

17. In Fig 6A, the authors show that the activation of bar-1, cwn-2 and mig-1 in Efm is hlh-26 dependent. It would be informative to show if these 3 genes are in the list of up-regulated genes in the RNAseq, and in addition, to show if they share the binding motifs of HLH-26.

18. In Fig 6C, the authors show the increasing BAR-1::GFP signal through Efm treatment in a hlh-26 dependent manner. It would be informative if the authors can show zoom in images with higher resolution to show the nuclear localization of BAR-1, since the activation of Wnt canonical signaling should induce the nuclear localization of BAR-1/β-catenin. The author may also show the cellular localization of the POP-1/TCF as the transcription factor regulated by Wnt signaling.

19. The authors state: "We provided evidence indicating that HLH-26-dependent immune genes were activated upon exposure to E. faecium to defend against S. enterica infection"

However, by the end of this manuscript it is still unclear which is the mechanism by which HLH-26 mediates E. faecium enhanced resistance against S. enterica.

Which of the defense effector genes upregulated in Fig. 4B contribute to E. faecium-enhanced resistance to S. enterica but not to resistance to single S. enterica infection? Meaning, which of those or other genes mediate probiotic-enhanced immunity in C. elegans?

So far, the authors present correlations between expression of genes downstream of HLH-26 and enhanced immunity; however, the only molecular players demonstrated to mediate the enhanced immunity are HLH-26 and BAR-1, and BAR-1 is a known mediator of the immune response in C. elegans and other organisms. Further, X et al already showed that bar-1-deficient C. elegans are compromised in their ability to mount a response to E. faecium exposure.

If the authors would shift the focus of their manuscript to presenting a function for a yet uncharacterized C. elegans transcription factor, then they may want to introduce the notion that HLH-26 seems to be a nematode-specific transcription factor. Hence, considering that the introduction focuses on the impact of probiotics on human health, the authors may want to elaborate on which lessons HLH-26 teach us that may be translatable to other organisms.

Minor points

1. Please add the line and page numbers to the manuscript and address the points by referring to the line and page numbers.

2. The author should specify the E. coli strains used in each experiment (e.g. OP50, HT115).

3. Fig 4B showed the qPCR data using bar graph. Box plot would be the better choice. Please go through all other figures/supplemental figures as well to change the bar graph to box plots.

4. In material and method, under statistical analysis, "ns" indicates non-significant. However, in all the Figure legends, non-significant is written as "NS". Check to make sure it is consistent.

5. Part of the conclusion of section 1 "and that HLH-26 was required for E. faecium-activated protection against S. enterica infection." Is demonstrated by experiments shown earlier in the section but not by the experiments showing that 7/10 e faecium-induced genes are hlh-26 dependent. Thus, decouple the conclusion from the QRT results and make it a general section conclusion or move this half of the conclusion upwards.

6. Because you are not looking at the protein, and much less its activity, using a promoter fusion, replace "…foci of HLH-26 activity" for "tissues in which hlh-26 is expressed"

7. In a similar note, the following is an overstatement: "The intestinal function of HLH-26 in protection against S. enterica infection was confirmed using strain MGH171, which allows intestine-specific RNAi." Because, although unlikely for a transcription factor, formally a an mRNA can be produced in the intestine, which will be abrogated by intestinal RNAi, whereas the protein may be exported and exert its critical function in a different tissue. In te absence of a protein function he authors need to continue using expressions as "suggest" or "is in line" instead of confirms.

8. Word missing here" Because BAR-1 plays a critical role in the response to ? and…"

9. Acknowledgment has a typo

10. The authors state: "The primer sequences are available upon request and are presented in S10

11. Table.". Please list all primers in table S10.

12. The authors state: "The animals were then visualized for bacterial load using a Leica M165 FC fluorescence stereomicroscope" However, no bacterial load experiments are presented here.

Rev. 2:

In this study, Sang and colleagues set out to identify mechanisms underlying E. faecium-mediated protection against Salmonella enterica infection using the model C. elegans. Through analyses of co-expressed genes during E. faecium exposure and predicted common binding motifs, the authors identified a requirement for hlh-26 in E. faecium-mediated protection against S. enterica infection. The authors further characterized the role of HLH-26 in this process through the use of transcriptional reporters, tissue-specific knockdown/rescue, mRNAseq, and gene enrichment analyses. They nicely showed that HLH-26 functions in the intestine to control this response and functions with the Wnt/BAR-1 pathway in this context. This manuscript logically follows the authors' previous publications on the regulation of intestinal immunity in C. elegans, and presents conclusions that are supported by the data presented.

Overall, I believe that the authors' have provided sufficient evidence that HLH-26 and the Wnt/BAR-1 pathway are involved in E. faecium-mediated protection in this S. enterica infection model. I have a few questions for the authors consideration:

1. Is HLH-30 activated in response to the proteolytic activity of SagA, a secreted protease from E. faecium, which was previously shown to control the protection of C. elegans against S. enterica (PMID: 27708039)? I think experiments to address this question should be included both to add mechanistic insight to the story, and also to reconcile the proposed mechanisms with what is already known about E. faecium-mediated protection against bacteria infection.

2. I am confused about the rationale for the experiments comparing the protection between heat-killed E. coli and live E. faecium. Is examining killed E. faecium on the protection against S. enterica a better experiment? In other words, do heat-killed E. faecium also provide resistance against S. enterica?

3. The authors demonstrate that hlh-26(ok1453) have reduced protection in the Efm-Se-Ec infection model. However, Figs. 2C, 2F, and 2H illustrate that the protection is not fully hlh-26-dependent and other factors may be important in E. faecium protection. Do ref-1;hlh-26 double mutants have complete abrogation of protection?

4. The title for Figure 2 does not properly represent the data shown.

5. Although the authors conclude that select innate immune genes are dependent on hlh-26 (Fig. 4B), the relevance of these innate immune genes in the protection against S. enterica is unclear. The authors could additionally show the difference in S. enterica CFUs between wild-type N2 and hlh-26(ok1453) after prophylaxis with E. faecium to demonstrate that these innate immune genes may be involved in the eradication of S. enterica. This experiment may not yield clean results, but perhaps is worth trying.

---

## [Decision Letter · Decision Letter 2]

7 Feb 2022

Dear Dr Aballay,

Thank you for submitting your revised Research Article entitled "The REF-1 family transcription factor HLH-26 controls the Wnt/BAR-1 in probiotic-mediated protection against gut infection" for publication in PLOS Biology. I have now obtained advice from the original reviewers and have discussed their comments with the Academic Editor. 

Based on the reviews (attached below), we will probably accept this manuscript for publication, provided you satisfactorily address the remaining points raised by Reviewer 1. Please also make sure to address the following data and other policy-related requests.

In addition, we would like to make a suggestion to improve the title:

"The transcription factor HLH-26 controls probiotic-mediated protection against Salmonella infection through upregulation of the Wnt/BAR-1 pathway"

We expect to receive your revised manuscript within two weeks. 

*Published Peer Review History*

*Early Version*

Sincerely,

Ines

--

Ines Alvarez-Garcia, PhD

Senior Editor,

ialvarez-garcia@plos.org,

PLOS Biology

DATA POLICY:

Many thanks for submitting the S1 Data file containing the data underlying all the graphs shown in the figures. However, we are missing the data from the following figures (if the data is already included in the manuscript, please indicate clearly where in the corresponding figure legends):

Fig. 3D (missing data from triple mutants); Fig. 4A, B; Fig. 5B, D, H; Fig. 6A, B, E, F, G (missing data from triple mutants); Fig. S2B; Fig. S6; Fig. S8A, B and Fig. S12

Please also ensure that figure legends in your manuscript include information on WHERE THE UNDERLYING DATA CAN BE FOUND.

In addition, you should make the RNA seq data deposited in Gene Expression Omnibus (GEO: GSE1780992) available at this stage, before the manuscript can enter production.

Reviewers' comments

Rev. 1:

The authors have satisfactorily addressed most of my concerns. Before publication, it would be necessary to address the remaining concerns listed below:

1) Explicitly state in the main text that the RNAseq data comes from a SINGLE replicate; therefore, several false negatives and false positives are expected.

2) Need to revise conclusions for hlh-26-dependent induction of some of the tested genes. For example, contrary to the conclusion, C04G6.5 in Fig. 4 is induced in the hlh-26 mutant to a similar extent as in the WT worms. The difference is that the max level induction is lower because hlh-26 affects the basal expression of this gene; however, the delta of expression Efaecium to Ecoli is still large in the absence of hlh-26. Therefore, it cannot be said that hlh-26 is required for C04G6.5 induction. What the data says is that HLH-26 is required for C04G6.5 basal expression and may subtly contribute to C04G6.5 induction. The same is the case for other genes including cwn-2 & R02E4.1 (Fig. 6) and C54D10.3 (Suppl. Fig. 6).

3) In response to the previous comment "Is BAR-1 overexpression sufficient to activate the immune genes upregulated in Fig. 4B? ", the authors responded: "… The immune genes listed in Fig. 4B are not predicted to have BAR-1 binding sites nor are they up-regulated by BAR-1 overexpression (Jackson et al. G3 (Bethesda). 2014;4(4):733-47)."

However, the fact that BAR-1 is not sufficient to activate the immune genes is not discussed in the manuscript. As a central player in the proposed model, BAR-1's insufficiency needs to be discussed.

Similarly, the previous comment on POP-1 was not fully addressed. POP-1 nuclear localization is controlled via its phosphorylation status. Therefore, even if pop-1 is not transcriptionally regulated, the localization of the POP-1 protein would be informative of its ON-OFF status. The authors need to at least discuss this point.

4) To reach a broader readership, replace N2 with WT in text and all figures

5) Describe exactly the number of base pairs cloned upstream of the ATG to generate the promoter fusion constructs and the primer sequences used. Do the same for the protein fusions; provide coordinates and primers used to make the constructs.

6) In Supplementary tables depicting GFP quantification, authors need to indicate the number of animals used to calculate the median/mean shown in the tables and disclose whether the depicted values and mean or medians. As of now, there is a single value per biological replicate.

7) Include the TF/s that bind to each gene in Table S3 worksheets named "28 TF controlled genes" & "5YF controlled genes".

Typos or similar

1. Across the manuscript, please change all qPCR instances for qRTPCR (because what is being measured is mRNAs)

2. Legend Fig.3A: the construct is referred to as HLH-26::GFP, but the image shows hlh-26P::GFP.

3. Title Fig 4: HLH-26 loss of function regulates an innate immune transcriptional program.

4. Lines 250-252: : "We tested verified by qRT-PCR the hlh-26-dependent regulation of ten genes involved in immune response (S6 Table) under four conditions: wild-type animals exposed to E. coli, wild type animals exposed to E. faecium, hlh-26(ok1453) animals exposed to E. coli, and hlh-26(ok1453) animals exposed to E. faecium.

5. Legend Fig. 5: Delete the Euro sign in "..expressing € empty.."

Rev. 2:

The authors have addressed my concerns from my initial review of the manuscript.

---

## [Editor Report · Decision Letter 3]

23 Feb 2022

Dear Dr Aballay,

On behalf of my colleagues and the Academic Editor, Ken Cadwell, I am pleased to say that we can in principle accept your Research Article entitled "The transcription factor HLH-26 controls probiotic-mediated protection against intestinal infection through upregulation of the Wnt/BAR-1 pathway" for publication in PLOS Biology, provided you address any remaining formatting and reporting issues. These will be detailed in an email that will follow this letter and that you will usually receive within 2-3 business days, during which time no action is required from you. Please note that we will not be able to formally accept your manuscript and schedule it for publication until you have any requested changes.

PRESS

Sincerely, 

Ines

--

Ines Alvarez-Garcia, PhD 

Senior Editor 

PLOS Biology
